# ADAPTIVE HIERARCHICAL CERTIFICATION FOR SEGMENTATION USING RANDOMIZED SMOOTHING

## ABSTRACT

Common certification methods operate on a flat pre-defined set of fine-grained classes. In this paper, however, we propose a novel, more general, and practical setting, namely adaptive hierarchical certification for image semantic segmentation. In this setting, the certification can be within a multi-level hierarchical label space composed of fine to coarse levels. Unlike classic methods where the certification would abstain for unstable components, our approach adaptively relaxes the certification to a coarser level within the hierarchy. This relaxation lowers the abstain rate whilst providing more certified semantically meaningful information. We mathematically formulate the problem setup and introduce, for the first time, an adaptive hierarchical certification algorithm for image semantic segmentation, that certifies image pixels within a hierarchy and prove the correctness of its guarantees. Since certified accuracy does not take the loss of information into account when traversing into a coarser hierarchy level, we introduce a novel evaluation paradigm for adaptive hierarchical certification, namely the certified information gain metric, which is proportional to the class granularity level. Our evaluation experiments on real-world challenging datasets such as Cityscapes and ACDC demonstrate that our adaptive algorithm achieves a higher certified information gain and a lower abstain rate compared to the current state-of-the-art certification method, as well as other non-adaptive versions of it.

## 1 INTRODUCTION

Image semantic segmentation is of paramount importance to many safety critical applications such as autonomous driving (Kaymak & Uçar, 2019; Zhang et al., 2016), medical imaging (Kayalibay et al., 2017; Guo et al., 2019; Pham et al., 2000), video surveillance (Cao et al., 2020) and object detection (Gidaris & Komodakis, 2015). However, ever since deep neural networks were shown to be inherently non-robust in the face of small adversarial perturbations (Szegedy et al., 2014), the risk of using them in such applications has become evident. Moreover, an arms race between new adversarial attacks and defenses has developed; which calls for the need of provably and certifiably robust defenses. Many certification techniques have been explored in the case of classification (Li et al., 2023), with the first recent effort in semantic segmentation (Fischer et al., 2021).

Certification for segmentation is a hard task since it requires certifying many components (i.e., pixels) simultaneously. The naive approach would be to certify each component to its top class within a radius, and then take the minimum as the overall certified radius of the image. This is problematic since a single unstable component could lead to a very small radius, or even abstain due to a single abstention. The state-of-the-art certification technique for semantic segmentation SEGCERTIFY (Fischer et al., 2021) relies on randomized smoothing (Lecuyer et al., 2019). It mitigates the many components issue by abstaining from unstable components and conservatively certifies the rest. While an unstable component implies that the model is not confident about a single top class, it often means that it fluctuates between classes that are semantically related. For example, if an unstable component fluctuates between *car* and *truck*, certifying it within a semantic hierarchy as *vehicle* would provide a more meaningful guarantee compared to abstaining.

We propose a novel hierarchical certification method for semantic segmentation, which adaptively certifies pixels within a multi-level hierarchical label space while preserving the same theoretical guarantees from Fischer et al. (2021). The hierarchy levels start from fine-grained labels to coarser

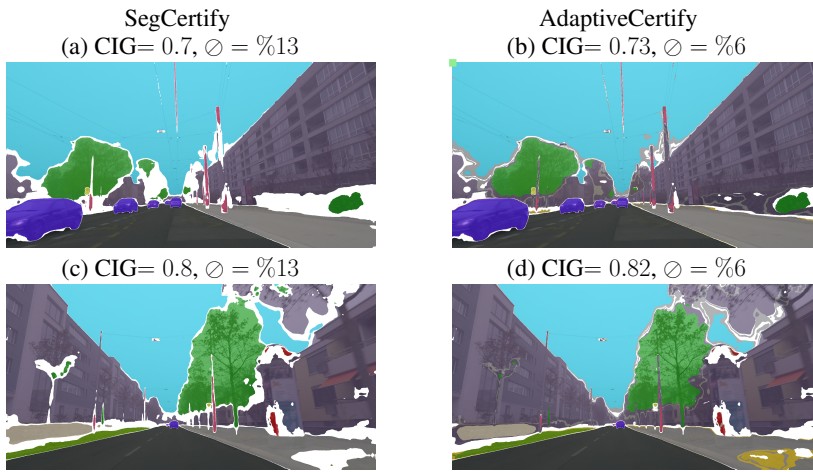

SegCertify
(a) CIG= 0.7, ⊘ = %13

AdaptiveCertify
(b) CIG= 0.73, ⊘ = %6

(c) CIG= 0.8, ⊘ = %13

(d) CIG= 0.82, ⊘ = %6

Figure 1: The certified segmentation outputs from SEGCERTIFY in (a) and (c), and ADAPTIVECER-TIFY in (b) and (d) with their corresponding certified information gain (CIG) and abstain rate %⊘. Our method provides more meaningful certified output in pixels the state-of-the-art abstains from (white pixels), with a much lower abstain rate, and higher certified information gain.

ones that group them. Our algorithm relies on finding unstable components within an image and relaxing their label granularity to be certified within a coarser level in a semantic hierarchy. Meanwhile, stable components can still be certified within a fine-grained level. As depicted in Figure 1, this relaxation of the label granularity lowers the abstain rate while providing more certified information to the end user compared to the state-of-the-art method.

To quantifiably evaluate our method, we propose a novel evaluation paradigm that accounts for the hierarchical label space, namely the certified information gain (CIG). The certified information gain is proportional to the granularity level of the certified label; a parent vertex (e.g., *vehicle*) has less information gain than its children (e.g., *car*, *truck*, *bus*, etc.) since it provides more general information, while leaf vertices have the most information gain (i.e., the most granular classes in the hierarchy). Our certified information gain metric is equivalent to certified accuracy if the defined hierarchy is flat.

**Main Contributions**. Our main contributions are the following: (i) We introduce the concept of adaptive hierarchical certification for image semantic segmentation by mathematically formulating the problem and its adaptation to a pre-defined class hierarchy, (ii) We propose ADAPTIVECERTIFY, the first adaptive hierarchical certification algorithm, which certifies the image pixels within different fine-to-coarse hierarchy levels, (iii) We employ a novel evaluation paradigm for adaptive hierarchical certification by introducing the certified information gain metric and (iv) we extensively evaluate our algorithm, showing that certifying each pixel within a multi-level classes hierarchy achieves a lower abstain rate and higher certified information gain than the current state-of-the-art certification method for segmentation. Our analysis further shows the generalization of ADAPTIVECERTIFY with respect to different noise levels and challenging datasets.

## 2 RELATED WORKS

**Certification.** The competition between attacks and defenses has resulted in a desire for certifiably robust approaches for verification and training (Li et al., 2023). Certification is proving that no adversarial sample can evade the model within a guaranteed range under certain conditions (Papernot et al., 2018). There are two major lines of certifiers, deterministic and probabilistic techniques.

Deterministic certification techniques such as SMT solvers (Pulina & Tacchella, 2010; 2012), Mixed-Integer Linear Programming (MILP) (Cheng et al., 2017; Dutta et al., 2018) or Extended Simplex Method (Katz et al., 2017) mostly work for small networks. To certify bigger networks, an

over-approximation of the network's output corresponding to input perturbations is required (Salman et al., 2019; Gowal et al., 2019), which underestimates the robustness.

Probabilistic methods work with models with added random noise: *smoothed models*. Currently, only probabilistic certification methods are scalable for large datasets and networks (Li et al., 2023). Randomized smoothing is a probabilistic approach introduced for the classification case against $l_p$ (Cohen et al., 2019) and non-$l_p$ threat models (Levine & Feizi, 2020). Beyond classification, it has been used in median output certification of regression models (Chiang et al., 2020), center-smoothing (Kumar & Goldstein, 2021) to certify networks with a pseudo-metric output space, and most relevant to our work, scaled to certify semantic segmentation models (Fischer et al., 2021). We expand on randomized smoothing and Fischer et al. (2021) in Section 3 to provide the necessary background for our work.

**Hierarchical Classification and Semantic Segmentation.** Hierarchical classification is the idea of classifying components within a hierarchical class taxonomy (Silla & Freitas, 2011). This taxonomy can be defined as a tree (Wu et al., 2005) or a Directed Acyclic Graph (DAG). A DAG means that a node can have multiple parent nodes, whereas trees only allow one. Hierarchical classifiers differ in how deep the classification in the hierarchy is performed. Some methods require the classification to be within the most fine-grained classes, namely mandatory leaf-node prediction (MLNP), whilst others can stop the classification at any level, namely non-mandatory leaf-node prediction (NMLNP) (Freitas & Carvalho, 2007). One way to deal with NMLNP is to set thresholds on the posteriors to determine which hierarchy level to classify at (Ceci & Malerba, 2007).

Hierarchical classification can be generalized to segmentation if the same hierarchical classification logic is done pixel-wise. In Li et al. (2022), a hierarchical semantic segmentation model is introduced that utilizes a hierarchy during training. NMLNP is by no means standard in current semantic segmentation work, even though it is, also from a practical perspective, useful for many down-stream tasks. Our certification for segmentation method follows an NMLNP approach: we can certify a pixel at a non-leaf node. Although we use some hierarchy-related concepts from previous works, our main focus is hierarchical certification for segmentation, not hierarchical segmentation, as we deal with the input segmentation model as a black-box.

## 3 PRELIMINARIES: RANDOMIZED SMOOTHING FOR SEGMENTATION

In this section, we provide an overview of the essential background and notations needed to understand randomized smoothing for classification and segmentation, which we build on when we introduce our adaptive method.

**Classification.** The core idea behind randomized smoothing (Cohen et al., 2019) is to construct a smoothed classifier $g$ from a base classifier $f$. The smoothed classifier $g$ returns the class the base classifier $f$ would return after adding as isotropic Gaussian noise to the input $x$. The smooth classifier is certifiably robust to $\ell_2$-perturbations within a certain radius. Formally, given a classifier $f : \mathbb{R}^m \mapsto \mathcal{Y}$ and Gaussian noise $\epsilon \backsim \mathcal{N}(0, \sigma^2 I)$, the smoothed classifier $g$ is defined as:

$$g(x) \coloneqq \arg\max_{a \in \mathcal{Y}} \mathbb{P}(f(x + \epsilon) = a) \tag{1}$$

In order to evaluate $g$ at a given input $x$ and compute the certification radius $R$, one cannot compute $g$ exactly for black-box classifiers due to its probability component. Cohen et al. (2019) use a Monte-Carlo sampling technique to approximate $g$ by drawing $n$ samples from $\epsilon \backsim \mathcal{N}(0, \sigma)$, evaluating $f(x + \epsilon)$ at each, and then using its outputs to estimate the top class and certification radius with a pre-set confidence of $1 - \alpha$, such that $\alpha \in [0, 1)$ is the type I error probability.

**Segmentation.** To adapt randomized smoothing to the segmentation case, Fischer et al. (2021) propose a mathematical formulation for the problem and introduce the scalable SEGCERTIFY algorithm to certify any segmentation model $f : \mathbb{R}^{3 \times N} \mapsto \mathcal{Y}^N$, such that $N$ is the number of components (i.e., pixels), and $\mathcal{Y}$ is the classes set. The direct application of randomized smoothing is done by applying the certification component-wise on the image. This is problematic since it gets affected dramatically by a single bad component by reporting a small radius or abstaining from certifying all components. SEGCERTIFY circumvents the bad components issue by side-stepping them by introducing a strict smooth segmentation model, that abstains from a component if the top class

probability is lower than a threshold $\tau \in [0, 1)$. The smooth model $g^\tau : \mathbb{R}^{3 \times N} \mapsto \hat{\mathcal{Y}}^N$ is defined as:

$$g_i^\tau(x) = \begin{cases} c_{A,i} & \text{if } \mathbb{P}_{\epsilon \sim \mathcal{N}(0,\sigma)}(f_i(x + \epsilon)) > \tau, \\ \oslash & \text{otherwise} \end{cases} \tag{2}$$

where $c_{A,i} = \arg\max_{c \in \mathcal{Y}} \mathbb{P}_{\epsilon \sim \mathcal{N}(0,\sigma)}(f(x + \epsilon) = c)$ and $\hat{\mathcal{Y}} = \mathcal{Y} \cup \{\oslash\}$ is the set of class labels combined with the abstain label. For all components where $g_i^\tau(x)$ commits to a class (does not abstain), the following theoretical guarantee holds:

**Theorem 1** (from (Fischer et al., 2021)). *Let $\mathcal{I}_x = \{i \mid g_i^\tau \neq \oslash, i \in 1, \ldots, N\}$ be the set of certified components indices in $x$. Then, for a perturbation $\delta \in \mathbb{R}^{N \times 3}$ with $||\delta||_2 < R := \sigma \Phi^{-1}(\tau)$, for all $i \in \mathcal{I}_x$: $g_i^\tau(x + \delta) = g_i^\tau(x)$.*

That is, if the smoothed model $g^\tau$ commits to a class, then it is certified with a confidence of $1 - \alpha$ to not change its output if the input perturbations are $l_2$-bounded by the radius $R$.

To estimate $g^\tau$, Fischer et al. (2021) employ a Monte-Carlo sampling technique in SEGCERTIFY to draw $n$ samples from $f(x + \epsilon)$ where $\epsilon \sim \mathcal{N}(0, \sigma)$, while keeping track of the class frequencies per pixel. With these frequencies, $p$-values are computed for hypothesis testing that either results in certification or abstain. Since there are $N$ tests performed at once, a multiple hypothesis scheme is used to bound the probability of type I error (FWER: family-wise error rate) to $\alpha$.

## 4 ADAPTIVE HIERARCHICAL CERTIFICATION

Enforcing to only certify components with a top-class probability $> \tau$ as previously described is a conservative requirement. While it mitigates the bad components effect on the certification radius by abstaining from them, those components are not necessarily "bad" in principle. Bad components have fluctuating classes due to the noise during sampling, which causes their null hypothesis to be accepted, and hence, are assigned $\oslash$. While this is a sign of the lack of the model's confidence in a single top class, it often means that the fluctuating classes are semantically related. For example, if sampled classes fluctuate between *rider* and *person*, this is semantically meaningful and can be certified under a grouping label *human*. This motivates the intuition behind our hierarchical certification approach, which can relax the sampling process to account for the existence of a hierarchy.

**Challenges:** To construct a certifier that adaptively groups the fluctuating components' outputs, there are three challenges to solve: (i) **Finding fluctuating components:** The question is: how do we find fluctuating or unstable components? Using the samples that are used in the statistical test would violate it since the final certificate should be drawn from i.i.d samples, (ii) **Adaptive sampling:** Assuming fluctuating components were defined, the adjustment of the sampling process to group semantically similar labels while working with a flat base model can be tricky. The challenge is to transform a model with flat, fine-grained labels into one whose output labels are part of a hierarchy while dealing with said model as a black-box, and (iii) **Evaluation:** Given a certifier that allows a component to commit to coarser classes, we need a fair comparison to other classical flat-hierarchy certification approaches (e.g., SEGCERTIFY). It is not fair to use the certified accuracy since it does not account for the information loss when grouping classes.

We construct a generalization of the smoothed model which operates on a flat-hierarchy of a pre-defined set of classes in Eq. 2 to formulate a hierarchical version of it. To recall in the definition, a smooth model $g^\tau$ certifies a component if it commits to a top class whose probability is $> \tau$, otherwise it abstains. The construction of $g^\tau$ deals with the model $f$ as a black-box, that is, by plugging in any different version of $f$, the same guarantees in Theorem 1 hold. We show the mathematical formulation of how we construct a hierarchical version of the smoothed model, and discuss how we overcome the challenges associated with it in this section.

### 4.1 HIERARCHICAL CERTIFICATION: FORMULATION

To define a hierarchical version of the smoothed model, we first replace the flat-hierarchy set of classes $\mathcal{Y}$ with a pre-defined class hierarchy graph $H = (\mathcal{V}, \mathcal{E})$, where the vertex set $\mathcal{V}$ contains semantic classes and the edge set $\mathcal{E}$ contains the relation on the vertices. Second, we define a hierarchical version of $f$, namely $f^H : \mathbb{R}^{N \times 3} \mapsto \mathcal{V}^N$, that maps the image components (pixels)

to the vertices $\mathcal{V}$. Third, we define a hierarchical smoothed model $c^{\tau,H} : \mathbb{R}^{N\times 3} \mapsto \hat{\mathcal{V}}^N$, such that $\hat{\mathcal{V}} = \mathcal{V} \cup \{\oslash\}$:

$$c_i^{\tau,H}(x) = \begin{cases} v_{A,i} & \text{if } \mathbb{P}_{\epsilon\sim\mathcal{N}(0,\sigma)}(f_i^H(x+\epsilon)) > \tau \\ \oslash, & \text{otherwise} \end{cases} \tag{3}$$

where $v_{A,i} = \arg\max_{v\in\mathcal{V}} \mathbb{P}_{\epsilon\sim\mathcal{N}(0,\sigma)}(f_i^H(x+\epsilon) = v)$. This certifier $c^{\tau,H}$ has three main novel components: the hierarchy graph $H$ (Section 4.2), the hierarchical function $f^H$ (Section 4.3) and the certification algorithm to compute $c^{\tau,H}$ (Section 4.4).

## 4.2 The Class Hierarchy Graph

We design the class hierarchy $H$ used by $c^{\tau,H}$ to capture the semantic relationship amongst the classes in $\mathcal{Y}$, as illustrated in Figure 2. $H$ is a pair $(\mathcal{V}, \mathcal{E})$ representing a DAG, where $\mathcal{V}$ is the set of vertices, and $\mathcal{E}$ is the set of edges representing the IS-A relationship among edges. We do not allow more than one parent for each vertex. An edge $e = (u, v) \in \mathcal{E}$ is defined as a pair of vertices $u$ and $v$, which entails that $u$ is a parent of $v$. The root vertex of the DAG denotes the most general class, *everything*, which we do not use. The hierarchy is divided into multiple levels $H_0, \ldots, H_3$, the more fine-grained the classes are, the lower the level. A hierarchy level is a set $H_l$ of the vertices falling within it. Leaf vertices $\mathcal{Y}$ are not parents of any other vertices but themselves. Essentially, $\mathcal{Y} = H_0$.

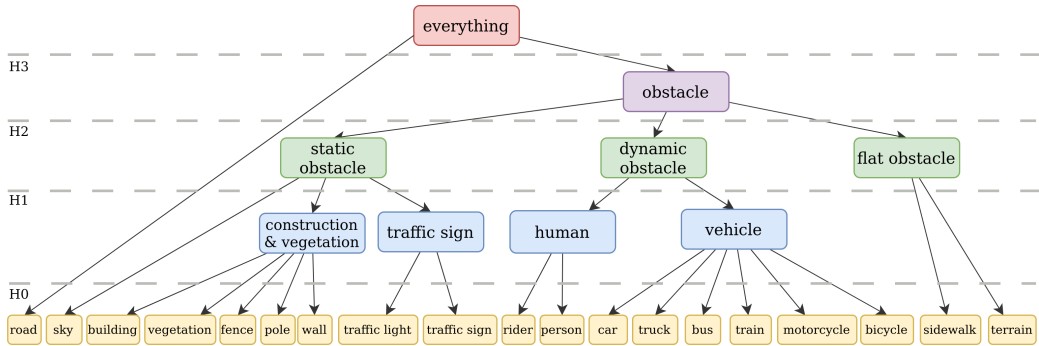

Figure 2: Illustration of a semantic hierarchy graph on Cityscapes classes.

## 4.3 Fluctuating components and adaptive sampling

In this part, we discuss how to solve two of the challenges concerning constructing an adaptive hierarchical certifier: defining fluctuating components without using the samples in the statistical test, and the adjustment of the sampling process to be adaptive.

We define the fluctuating components by an independent set of samples from those used in the hypothesis test. We first draw initial $n_0$ posterior samples per component from the segmentation head of $f$, defined as $f_{\text{seg}} : \mathbb{R}^{N\times 3} \mapsto [0,1]^{N\times|\mathcal{Y}|}$. We then look at the top two classes' mean posterior difference. The smaller the difference, the coarser the hierarchy level the component is assigned to. These steps are outlined in Algorithm 1 describing GetComponentLevels, which finds the hierarchy level index for every component.

We invoke SamplePosteriors to draw initial $n_0$ samples from $f_{\text{seg}}(x+\epsilon)$, $\epsilon \sim \mathcal{N}(0,\sigma)$. This method retrieves $n_0$ posteriors per component: $Ps_1^0, \ldots, Ps_N^0$, such that $Ps_i^0$ is a set of $n_0$ posterior vectors $\in [0,1]^{|\mathcal{Y}|}$ for the $i$-th component. Then, for every component $i$, we get the mean of its $n_0$ posteriors $P_i^0$, and calculate the posterior difference $\Delta P_i$ between the top two classes, indexed by $\hat{c}_{A_i}$ and $\hat{c}_{B_i}$. We use thresholds to determine its hierarchy level index $l$ by invoking a threshold function $T_{\text{thresh}}$. Given a hierarchy with $L$ levels, the threshold function $T_{\text{thresh}}$ is defined as:

$$T_{\text{thresh}}(\Delta P_i) = \underset{l\in\{0,\ldots,L-1\}}{\arg\min} \ t_l, \text{ s.t. } t_l < \Delta P_i \tag{4}$$

with $t_l \in [0,1]$. $T_{\text{thresh}}$ returns the index of the most fine-grained hierarchy level the component can be assigned to based on the pre-set thresholds $t_0 > t_1 > \ldots > t_{L-1}$.

---

**Algorithm 1** GETCOMPONENTLEVELS: algorithm to map components to hierarchy levels

---

**function** GETCOMPONENTLEVELS($f, x, n_0, \sigma$)
    $Ps_1^0, \ldots, Ps_N^0 \leftarrow$ SAMPLEPOSTERIORS($f, x, n_0, \sigma$)
    **for** i $\leftarrow 1, \ldots,$ N **do**
        $P_i^0 \leftarrow$ mean $Ps_i^0$
        $\hat{c}_{A_i}, \hat{c}_{B_i} \leftarrow$ top two class indices $P_i^0$
        $\Delta P_i \leftarrow P_i^0[\hat{c}_{A_i}] - P_i^0[\hat{c}_{B_i}]$
        $l_i \leftarrow T_{\text{thresh}}(\Delta P_i)$
    **return** $(l_1, \ldots, l_N), (\hat{c}_{A_1}, \ldots, \hat{c}_{A_N})$

---

Now that we know which level index every component is mapped to, we can define $f^H$, which takes an image $x$ and does a pixel-wise mapping to vertices $\hat{v}$ within every component's assigned hierarchy level $H_{l_i}$. Mathematically, we define the predicted label $\hat{v}_i$ component $i$ as:

$$f_i^H(x) = K(f_i(x), l_i) = \hat{v}_i \iff \exists_{\hat{v}_i, u_1, \ldots, u_p, \hat{y}_i}(\{(\hat{v}_i, u_1), \ldots, (u_p, \hat{y}_i)\} \subseteq \mathcal{E}) \wedge (\hat{v}_i \in H_{l_i}) \quad (5)$$

such that there is a path from the parent vertex $\hat{v}_i$ that belongs to the hierarchy level $H_{l_i}$ to the predicted leaf $\hat{y}_i = f_i(x)$. For example, $K(bus, 0) = bus$, $K(bus, 1) = vehicle$ and $K(bus, 2) = dynamic\ obstacle$. Constructing a smoothed version of $f^H$, namely $c_i^{\tau, H}$, is now equivalent to the hierarchical certifier we formulated earlier in Eq. 3.

Evaluating $c^{\tau, H}$ requires a sampling scheme over $f^H$ to get the top vertex frequencies $\text{cnts}_1, \ldots, \text{cnts}_N$ as outlined in Algorithm 2. The sampling of $f^H(x + \epsilon)$ such that $\epsilon \curvearrowleft \mathcal{N}(0, \sigma)$ is a form of adaptive sampling over $f$. For the $i^{\text{th}}$ component with level $l_i$, $f^H$ is invoked on $x + \epsilon$, which in its definition invokes $f(x + \epsilon)$ to output a flat segmentation label map $\hat{y}$, whose component $\hat{y}_i$ is mapped to its parent vertex $\hat{v}_i$ in $H_{l_i}$ using the function $K$ as in Eq. 5.

---

**Algorithm 2** HSAMPLE: algorithm to adaptively sample

---

**function** HSAMPLE($f, K, (l_1, \ldots, l_N), x, n, \sigma$)
    $\text{cnts}_1, \ldots, \text{cnts}_N \leftarrow$ initialize each to a zero vector of size $|\mathcal{V}|$
    draw random noise $\epsilon \curvearrowleft \mathcal{N}(0, \sigma)$
    **for** $j \leftarrow 1, \ldots, n$ **do**
        $\hat{y} = f(x + \epsilon)$ $\quad\quad\quad$ } mathematically
        **for** $i \leftarrow 1, \ldots, N$ **do** $\quad$ equivalent
            $\hat{v}_i \leftarrow K(\hat{y}[i], l[i])$ $\quad$ to calling
            $\text{cnts}_i[\hat{v}_i]\ += 1$ $\quad\quad \equiv f^H(x + \epsilon)$
    **return** $\text{cnts}_1, \ldots, \text{cnts}_N$

---

### 4.4 OUR ALGORITHM: ADAPTIVECERTIFY

Putting it all together, we now introduce ADAPTIVECERTIFY 3, which overcomes the challenges of defining the fluctuating components and employing an adaptive sampling scheme to certify an input segmentation model $f$ given a hierarchy $H$. Our certification algorithm approximates the smoothed model $c^{\tau, H}$ following a similar approach by Fischer et al. (2021).

---

**Algorithm 3** ADAPTIVECERTIFY: algorithm to hierarchically certify and predict

---

**function** ADAPTIVECERTIFY($f, K, \sigma, x, n, n_0, \tau, \alpha$)
    $(l_1, \ldots, l_N), (\hat{c}_{A_1}, \ldots, \hat{c}_{A_N}) \leftarrow$ GETCOMPONENTLEVELS($f, x, n_0, \sigma$)
    $\hat{v}_1, \ldots, \hat{v}_N \leftarrow$ Use $K$ and $l_i$ to get parent vertices of $\hat{c}_{A_1}, \ldots, \hat{c}_{A_N}$
    $\text{cnts}_1, \ldots, \text{cnts}_N \leftarrow$ HSAMPLE($f, K, (l_1, \ldots, l_N)$, x, n, $\sigma$)
    $pv_1, \ldots, pv_N \leftarrow$ BINPVALUE($(\hat{v}_1, \ldots, \hat{v}_N), (\text{cnts}_1, \ldots, \text{cnts}_N), \tau$)
    $\hat{v}_1, \ldots, \hat{v}_N \leftarrow$ HYPOTHESESTESTING($\alpha, \oslash, (pv_1, \ldots, pv_N), (\hat{v}_1, \ldots, \hat{v}_N)$)
    $R \leftarrow \sigma \Phi^{-1}(\tau)$
    **return** $\hat{v}_1, \ldots, \hat{v}_N, R$

---

On a high level, ADAPTIVECERTIFY consists of three parts: (i) mapping components to hierarchy level indices by invoking GETCOMPONENTLEVELS, (ii) adaptively sampling to estimate the smoothed model $c^{\tau,H}$ by invoking HSAMPLE, and (iii) employing multiple hypothesis testing via HYPOTHESESTESTING to either certify a component or assign $\oslash$ to it. To avoid invalidating our hypotheses test, we use the initial set of $n_0$ independent samples drawn in GETCOMPONENTLEVELS to both decide on the assigned component levels indices $l_1, \ldots, l_N$, as well as the top class indices $\hat{c}_{A_1}, \ldots, \hat{c}_{A_N}$. Since those classes are in $\mathcal{Y}$ since they come from the flat model $f$, we transform them using the mapping function $K$ the levels to get their corresponding parent vertices in the hierarchy: $\hat{v}_1, \ldots, \hat{v}_N$. These vertices are used to decide on the top vertex class, while the counts drawn from the adaptive sampling function HSAMPLE are used in the hypothesis testing. With these counts, we perform a one-sided binomial test on every component to retrieve its $p$-value, assuming that the null hypothesis is that the top vertex class probability is $< \tau$. Then, we apply multiple hypothesis testing to reject (certify) or accept (abstain by overwriting $\hat{v}_i$ with $\oslash$) from components while maintaining an overall type I error probability of $\alpha$.

We now show the soundness of ADAPTIVECERTIFY using Theorem 1. That is, if ADAPTIVECERTIFY returns a class $\hat{v}_i \neq \oslash$, then with probability $1 - \alpha$, the vertex class is certified within a radius of $R := \sigma\Phi^{-1}(\tau)$, same as in Fischer et al. (2021).

**Proposition 1** (Similar to (Fischer et al., 2021)). *Let $\hat{v}_1, \ldots, \hat{v}_N$ be the output of* ADAPTIVECERTIFY *given an input image $x$ and $\hat{I}_x := \{\oslash \mid \hat{v}_i \neq \oslash\}$ be the set of non-abstain components indices in $x$. Then with probability at least $1 - \alpha$, $\hat{I}_x \subseteq I_x$ such that $I_x$ denotes the theoritical non-abstain indices previously defined in Theorem 1 by replacing $g^\tau$ with our smoothed model $c^{\tau,H}$. Then, $\forall i \in \hat{I}_x$, $\hat{v}_i = c_i^{\tau,H}(x) = c_i^{\tau,H}(x + \delta)$ for an $l_2$ bounded noise $||\delta||_2 \leq R$.*

*Proof.* With probability $\alpha$, a type I error would result in $i \in \hat{I}_x \setminus I_x$. However, since $\alpha$ is bounded by HYPOTHESESTESTING, then with probability at least $1 - \alpha$, $\hat{I}_x \subseteq I_x$. $\square$

### 4.5 PROPERTIES OF ADAPTIVECERTIFY

The hierarchical nature of ADAPTIVECERTIFY means that instead of abstaining for unstable components, it relaxes the certificate to a coarser hierarchy level. While not always successful, this increases the chances for certification to succeed on a higher level. The abstaining can still occur on any hierarchy level, and it has two reasons: the top vertex probability is $\leq \tau$, and by definition $c^{\tau,H}$ would abstain, or it is a Type II error in ADAPTIVECERTIFY.

ADAPTIVECERTIFY guarantees that the abstain rate is always less than or equal to a non-adaptive flat-hierarchy version (e.g., SEGCERTIFY). If our algorithm only uses level $H_0$ to all components, the abstain rate will be equal to a non-adaptive version. So, since some components are assigned to a coarser level, their $p$-values can only decrease, which can only decrease the abstain rate.

By adapting the thresholds in $T_{\text{thresh}}$ and the hierarchy definition, one can influence the hierarchy levels assigned to the components. Strict thresholds or coarser hierarchies would allow most components to fall within coarse levels. This is a parameterized part of our algorithm that can be adjusted based on the application preferences, trading off the certification rate versus the certified information gain, which we detail in App A.1.

### 4.6 EVALUATION PARADIGM: CERTIFIED INFORMATION GAIN (CIG)

As mentioned previously, certified accuracy does not take the loss of information into account when traversing into coarser hierarchies, it would be trivial to maximize certified robustness by assigning all components to the topmost level. We therefore define a new certified information gain (CIG) metric that is proportional to the class granularity level. That is, a pixel gets maximum CIG if certified within the most fine-grained level $H_0$, and it decreases the higher the level is.

Formally, the certified information gain metric CIG for a component $i$, its certified vertex $\hat{v}_i$, its ground truth flat label $y_i$, and its hierarchy level $L_i$ is defined as:

$$\text{CIG}(\hat{v}_i, y_i, L_i) = \begin{cases} \log(|\mathcal{Y}|) - \log(\text{generality}(v_i)) & \hat{v}_i = K(y_i, L_i) \\ 0 & \text{otherwise.} \end{cases} \tag{6}$$

$|\mathcal{Y}|$ is the number of leaves (i.e., the number of the most fine-grained classes), and $\mathrm{generality}(v_i)$ is defined as the number of leaf vertices that are reachable by $v_i$. Assuming certification succeeds, CIG is maximized when $\hat{v}_i$ is a leaf vertex since $\mathrm{CIG}_i(x) = \log(|\mathcal{Y}|) - 0$. We normalize CIG by $\log(|\mathcal{Y}|)$, which results in a score between 0 and 1 and reduces to certified accuracy for non-adaptive algorithms that only consider $H_0$.

## 5 RESULTS

We evaluate ADAPTIVECERTIFY in a series of experiments to show its performance against the current state-of-the-art, and illustrate its hierarchical nature. We use two segmentation datasets: Cityscapes (Cordts et al., 2016) and the Adverse Conditions Dataset with Correspondences (ACDC) (Sakaridis et al., 2021). Cityscapes contains images ($1024 \times 2048$ px) of scenes across different cities. ACDC is a challenging traffic scene dataset with images across four adverse visual conditions: snow, rain, fog, and nighttime. It uses the same 19 classes as Cityscapes ($H_0$ in Figure 2). Our implementation can be found in the supplementary material and will be published with the paper.

We use HrNetV2 (Sun et al., 2019; Wang et al., 2019) as the uncertified base model, trained on Gaussian noise with $\sigma = 0.25$ by Fischer et al. (2021). The inference is invoked on images with their original dimension without scaling. We use the hierarchy defined in Figure 2 and $n_0 = 10$, $n = 100, \tau = 0.75$ throughout experiments unless stated otherwise. We use different parameters for the threshold function $T_{\mathrm{thresh}}$ (Eq 4) for ADAPTIVECERTIFY on both datasets which we denote as a 3-tuple $(t_2, t_1, t_0)$. The choice of the 3-tuple is a consequence of the 4-level hierarchy. The values used are $(0, 0, 0.25)$ for Cityscapes and $(0, 0.05, 0.3)$ for ACDC, which we found via a grid search that maximizes the certified information gain, as detailed in App. A.2.

Table 1: Certified segmentation results for 200 images from each dataset. CIG stands for per-pixel certified information gain, and %⊘ is the abstain rate. Our method AdaptiveCertify and SegCertify use $n_0 = 10, \alpha = 0.0001$, and their CIG is certified within a radius $R$ at different noise levels $\sigma$, thresholds $\tau$, and number of samples $n$.

| | $\sigma$ | $R$ | Cityscapes | | ACDC | |
| --- | --- | --- | --- | --- | --- | --- |
| | | | CIG | %⊘ | CIG | %⊘ |
| Uncertified HrNet | - | - | 0.91 | - | 0.55 | - |
| SEGCERTIFY | 0.25 | 0.17 | 0.87 | 11 | 0.53 | 34 |
| | 0.33 | 0.22 | 0.77 | 21 | 0.43 | 44 |
| | 0.50 | 0.34 | 0.35 | 41 | 0.17 | 38 |
| $n = 100,$ $\tau = 0.75$ ADAPTIVECERTIFY | 0.25 | 0.17 | **0.88** (↑ 0.01) | **9** (↓ 2%) | **0.54** (↑ 0.01) | **28** (↓ 6%) |
| | 0.33 | 0.22 | **0.79** (↑ 0.02) | **17** (↓ 3%) | **0.45** (↑ 0.02) | **35** (↓ 8%) |
| | 0.50 | 0.34 | **0.38** (↑ 0.03) | **30** (↓ 11%) | **0.20** (↑ 0.03) | **29** (↓ 9%) |
| SEGCERTIFY | 0.25 | 0.41 | 0.84 | 14 | 0.49 | 41 |
| | 0.33 | 0.52 | 0.72 | 27 | 0.38 | 53 |
| | 0.50 | 0.82 | 0.31 | 50 | 0.15 | 47 |
| $n = 500,$ $\tau = 0.95$ ADAPTIVECERTIFY | 0.25 | 0.41 | **0.85** (↑ 0.01) | **12** (↓ 2%) | **0.51** (↑ 0.02) | **36** (↓ 5%) |
| | 0.33 | 0.52 | **0.73** (↑ 0.01) | **24** (↓ 3%) | **0.40** (↑ 0.02) | **46** (↓ 7%) |
| | 0.50 | 0.82 | **0.34** (↑ 0.03) | **41** (↓ 10%) | **0.18** (↑ 0.03) | **38** (↓ 8%) |

We first investigate the overall performance of ADAPTIVECERTIFY against the current state-of-the-art SEGCERTIFY across different noise levels $\sigma$ and number of samples $n$. We show the results on both datasets in Table 1. On a high level, ADAPTIVECERTIFY consistently has a higher certified information gain and lower abstention rate than SEGCERTIFY. Although increasing the noise level $\sigma$ degrades the performance in both algorithms, ADAPTIVECERTIFY abstains much less than SEGCERTIFY, while maintaining a higher certified information gain, at higher noise levels. We look more into the effect of increasing the noise level in App. A.2.

We now take a closer look at the performance of ADAPTIVECERTIFY versus SEGCERTIFY by varying the number of samples. We show in Figure 3 that ADAPTIVECERTIFY consistently outperforms SEGCERTIFY in terms of the certification rate and CIG.

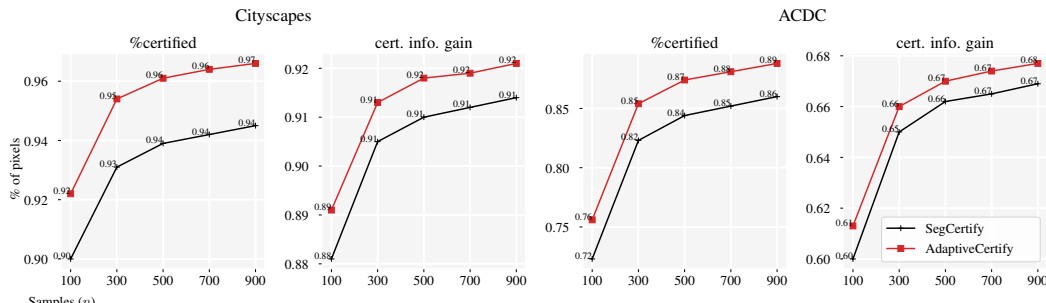

Figure 3: %certified (mean per-pixel certification rate) and cert. info. gain (mean per-pixel certified information gain) versus the number of samples $n$ on Cityscapes and ACDC.

To illustrate the hierarchical nature of ADAPTIVECERTIFY, we inspect the pixel distribution across hierarchy levels. We show results in Figure 4. For ADAPTIVECERTIFY, most pixels are certified at $H_0$, and only a small percentage of pixels fall under coarser hierarchy levels. ACDC has more pixels in coarser levels ($H_1$ and $H_2$) than Cityscapes ($H_1$). This is because ACDC is a more challenging dataset, leading to more fluctuating components, which our algorithm assigns to coarser levels instead of abstaining. ADAPTIVECERITFY correctly certifies almost the same percentage of pixels in $H_0$ as SEGCERTIFY, and due to our relaxation, we certify an additional $3\%$ and $6\%$ of the pixels in Cityscapes and ACDC at coarser levels.

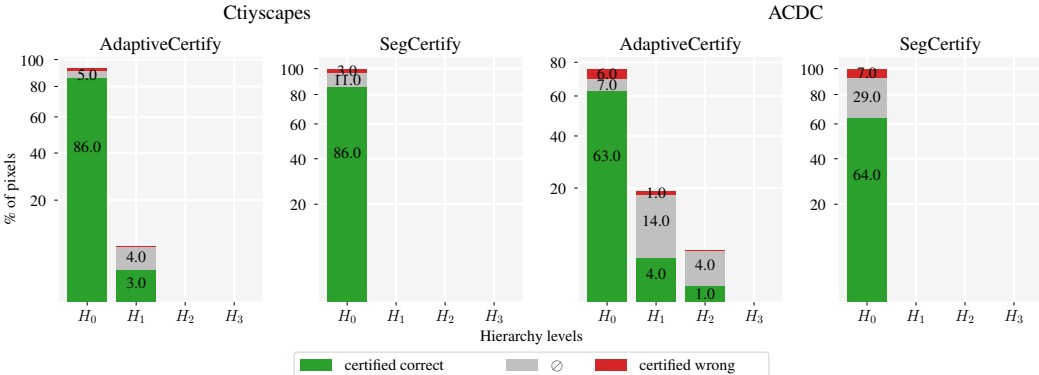

Figure 4: The performance in terms of percentage of abstain ($\oslash$), certified wrong (certified label doesn't match the ground truth) and certified correct (certified label matches the ground truth) under hierarchy levels from Figure 2. Note that SEGCERTIFY, by definition, only uses $H_0$.

## 6 CONCLUSION

In this paper, we introduced adaptive hierarchical semantic segmentation. We mathematically formulated the problem and proposed a method that solves three main challenges: finding bad components, relaxing the sampling process for those components to certify them within a coarser level of a semantic hierarchy, and evaluating the results using the certified information gain metric. We proposed our novel method ADAPTIVECERTIFY for adaptive hierarchical certification for image semantic segmentation, which solves these challenges. Instead of abstaining for unstable components, ADAPTIVECERTIFY relaxes the certification to a coarser hierarchy level. It guarantees an abstain rate less than or equal to non-adaptive versions while maintaining a higher CIG across different noise levels and number of samples. The formulation of our hierarchical certification method is general and can adopt any hierarchy, which allows for adaptation to different tasks.

**Reproducibility Statement.** In this paper, we describe our novel algorithm ADAPTIVECERTIFY on multiple levels: (1) We describe the algorithm in detail in Section 4, supported by mathematical definitions throughout the section. (2) We discuss its implementation using high-level pseudo-code in Algorithms 1, 2, and 3. (3) We are sharing an anonymous link to our code for ADAPTIVECERTIFY and the experiment scripts with the reviewers, which we will publish in a formal repository for public access once the paper is accepted. The datasets and hyper-parameters used for the experiments are specified in the beginning of Section 5, and we provide details on how to set the thresholds of $T_{\mathrm{thresh}}$ in App. A.2.

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

# A APPENDIX

## A.1 HIERARCHY LEVELS CERTIFIED INFO. GAIN (CIG) VS. CERTIFICATION RATE TRADEOFF

We want to analyze the influence different hierarchy levels have on abstention rate and certified information gain. To this end, we create non-adaptive versions of our algorithm for each hierarchy $H_i$. We denote a non-adaptive version of our algorithm that only uses a single hierarchy level $H_i$ as *Non-adaptive-$H_i$*. Since SEGCERTIFY is non-adaptive and restricted to $H_0$, it is equivalent to *Non-adaptive-$H_0$*. Figures 5 and 6 show the results of these experiments for different numbers of samples n and noise levels $\sigma$ on both datasets. ADAPTIVECERTIFY consistently has a higher certification rate and certified information gain than SEGCERTIFY. In Figure 5, both algorithms' performance increases with increasing $n$ since having more samples leads to more evidence to reject the null hypothesis, and hence, certifying more components. In Figure 6, the performance decreases by increasing $\sigma$ for both algorithms. In both figures, for the none-adaptive algorithms, a coarser level leads to a higher certification rate but lower certified information gain. This makes sense, as relaxing the certification to coarse levels only makes the certification easier since the labels become less granular, it comes at the cost of a decreased certified information gain since the certified vertices labels are more general. Our algorithm finds a balance between all hierarchies: if a component is found to be stable, it will be certified within a fine-grained level (e.g., $H_0$), maximizing the information. Whereas an unstable component will be certified at coarser levels instead of abstaining, which makes it outperform all other non-adaptive setups considering the certification rate and certified information gain trade-off.

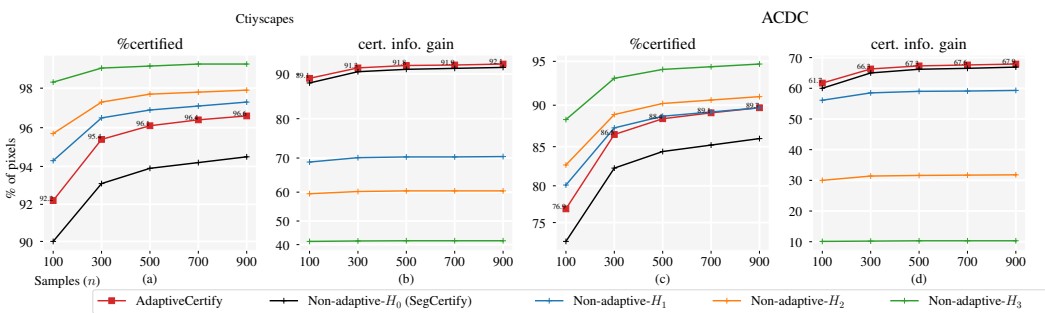

Figure 5: %certified (mean per-pixel certification rate) and cert. info. gain (mean per-pixel certified information gain) versus the number of samples $n$ ($n_0 = 10, n = 100, \tau = 0.75$) on Cityscapes and ACDC.

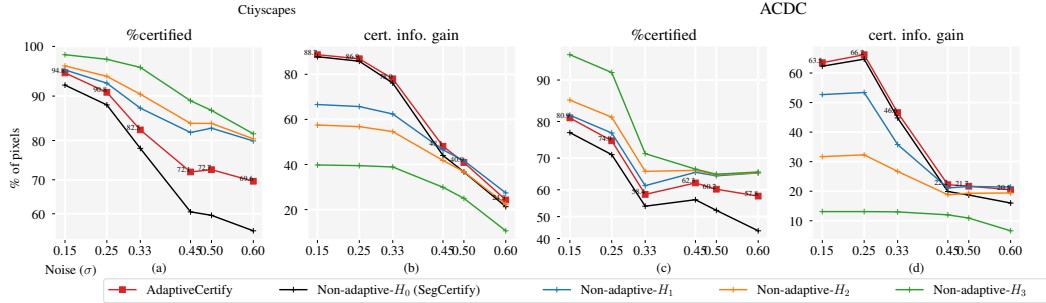

Figure 6: %certified (mean per-pixel certification rate) and cert. info. gain (mean per-pixel certified information gain) versus the noise levels $\sigma$ ($n_0 = 10, n = 100, \tau = 0.75$) on Cityscapes and ACDC.

## A.2 THRESHOLD SEARCH

We use grid search over different threshold function parameters $T_{\text{thresh}}$ and measure the performance across different $n$ in Figure 7 and noise level $\sigma$ in Figure 8. We find that the best thresholds for both datasets that give the highest mean certified information gain compared to the rest are $(0, 0, 0.25)$ and $(0, 0.05, 0.3)$ on Cityscapes and ACDC, respectively.

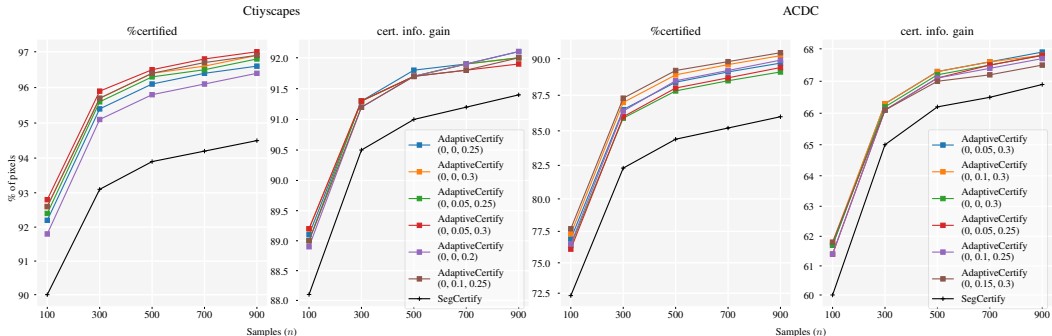

Figure 7: An extended version of Figure 3. The performance of SEGCERTIFY (black line) against multiple versions of ADAPTIVECERTIFY by varying the threshold function parameters $T_{\text{thresh}}$ on both datasets. The legend is ordered in a descending order of the performance in terms of the mean certified information gain across the number of samples $n$. This is a result of a grid search over 63 threshold functions, but we are plotting only some of them for clarity.

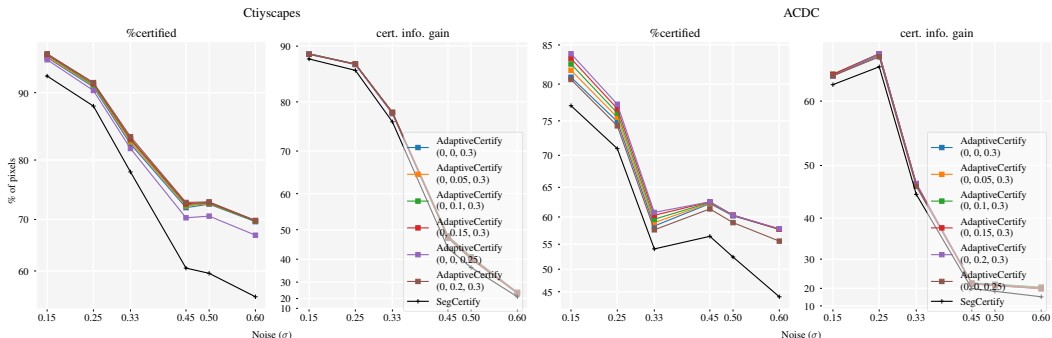

Figure 8: An extended version of Figure 6. The performance of SEGCERTIFY (black line) against multiple versions of ADAPTIVECERTIFY by varying the threshold function parameters $T_{\text{thresh}}$ on both datasets. The legend is ordered in a descending order of the performance in terms of the mean certified information gain across different noise levels $\sigma$. This is a result of a grid search over 63 threshold functions, but we are plotting only some of them for clarity.

## A.3 VISUAL RESULTS

We sample images from both datasets Cityscapes and ACDC to visually evaluate the performance of ADAPTIVECERTIFY (ours) against SEGCERTIFY. In Figure 9, we show selected examples that resemble a significant improvement from ADAPTIVECERITIFY against the baseline in terms of the certified information gain and abstain rate. In Figures 10 and 11, we show randomly picked samples to evaluate the average performance of our method.

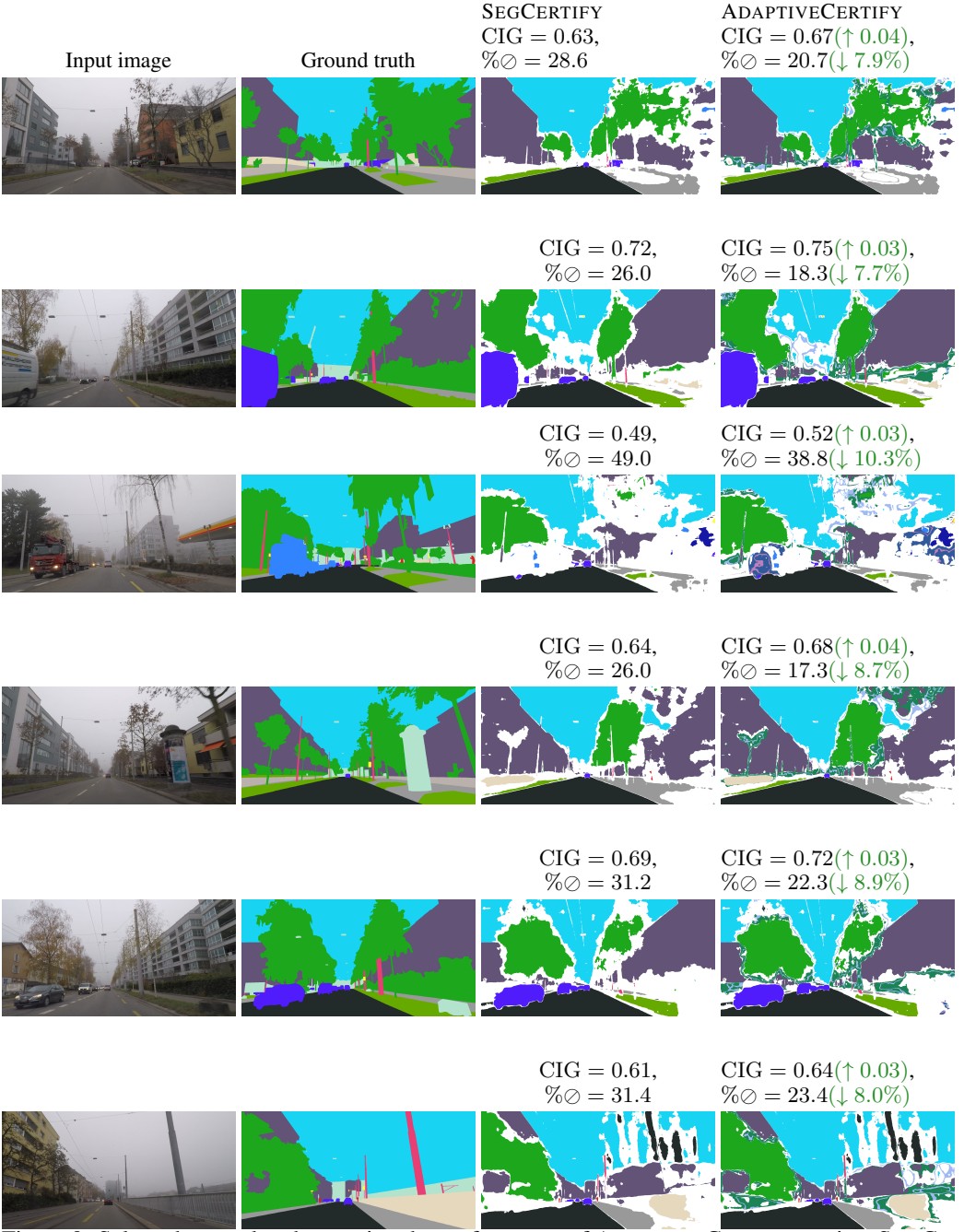

Figure 9: Selected examples showcasing the performance of ADAPTIVECERTIFY against SEGCERTIFY. The certified information gain is consistently higher by at least 0.03 while the abstain rate is lower by at least 6.8%.

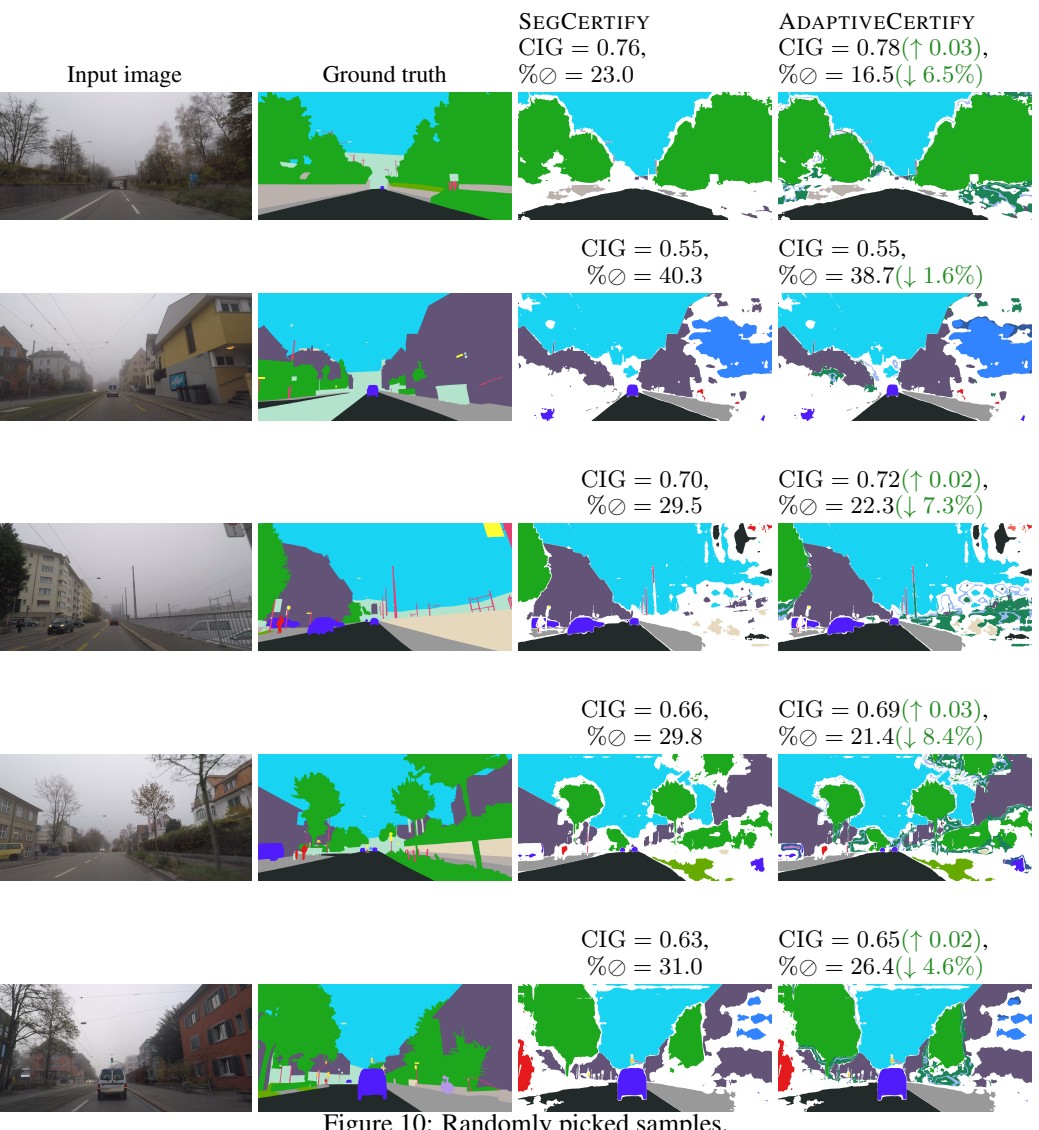

Figure 10: Randomly picked samples.

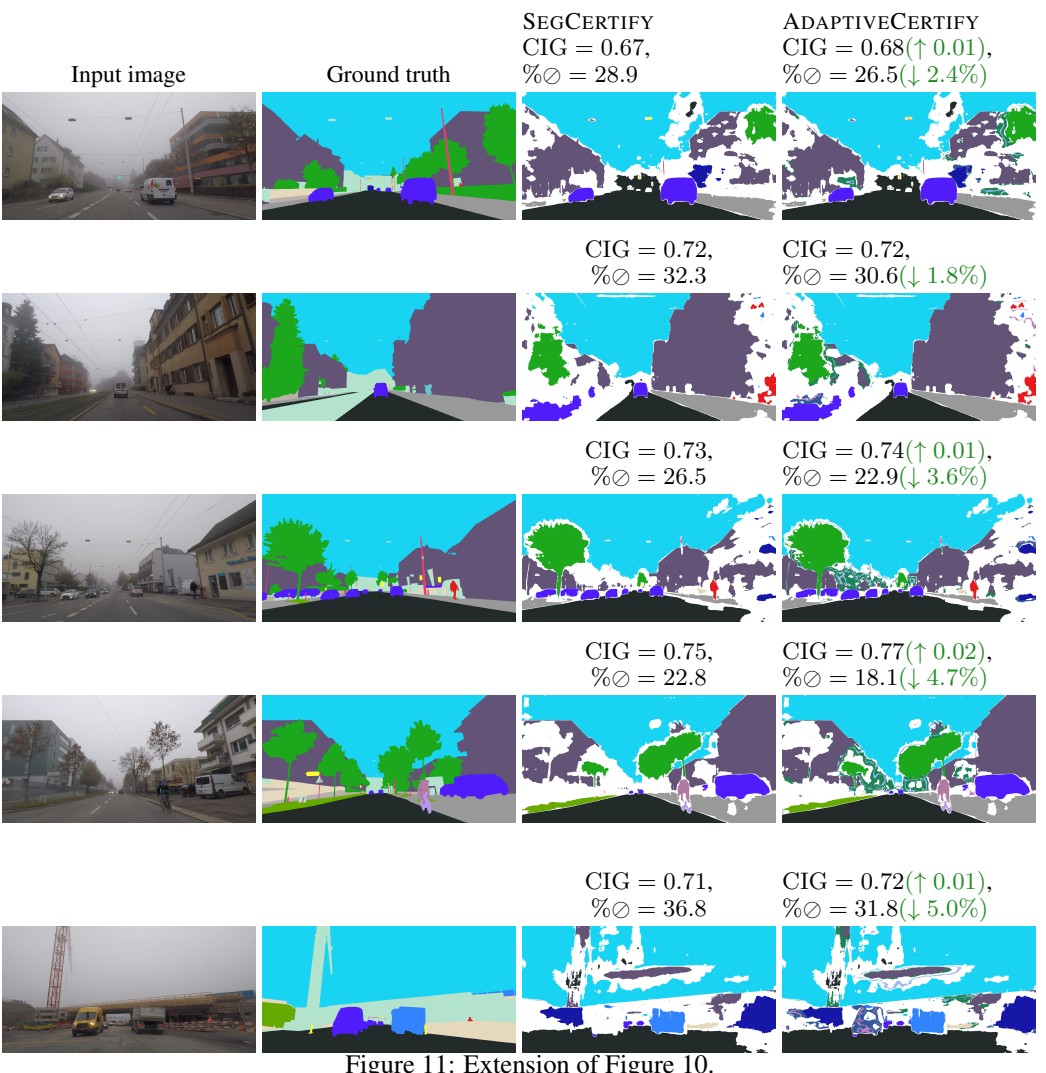

Figure 11: Extension of Figure 10.

