# OpenReview forum: "Adaptive Hierarchical Certification for Semantic Segmentation using Randomized Smoothing"
_ICLR.cc/2024/Conference — Submitted to ICLR 2024_

### Official Review · Reviewer_riuJ · 2023-10-30

**Soundness:** 2 fair
**Presentation:** 3 good
**Contribution:** 3 good
**Rating:** 3
**Confidence:** 4

**Summary:**

This paper aims to present a new certification method for the semanitc segmentation task. Previous relevant works mostly aim to solve the classification task. This paper is among the first to explore a new certification method, namely adaptive hierarchical certification and design a new evaluation metric. The authors clearly explain the proposed method and conduct a series of experiments to show that the proposed adaptive hierarchical certification performs better than previous works.

**Strengths:**

- The novelty of this paper is clear. Introducing a hierarchical certification method is more suitable for the semantic segmentation task, which needs to predict a class for each pixel.

- The presentation of this paper is good. The authors clearly explain the background of the proposed approach and then clearly describe the method.

- Experiments show that the proposed approach receives good results in terms of the proposed CIG metric.

**Weaknesses:**

- It seems that Theorem 1 is originally from (Fischer et al., 2021), not sure why put it in the main paper.

- It is glad to see that the authors proposed to use the class hierarchy graph to do certification, which has never been proposed in previous works as far as I know. However, a problem of introducing class hierarchy is that when applied to a new dataset, for example, the ADE20k dataset, a new class hierarchy graph should be prepared. Not sure how to make the proposed method universal to different segmentation datasets.

- Fromt the paper, it seems that the proposed method only use two datasets, i.e., Cityscapes and ACDC, to evaluate the proposed method. As these two datasets, as mentioned in the paper, are composed of only 19 classes, two questions consequently come:

  - First, more datasets should be used to evaluate the performance of the proposed method.
  - Second, the Cityscapes dataset only contain 19 classes. I think the authors should do some experiments on some dataset with more classes. A proper one might be the ADE20K dataset, which has more than 100 semantic categories. This would definitely verify the effectiveness of the adaptive hierarchical certification method.

- I also think some segmentation results should be visualized to see in which cases the proposed adaptive hierarchical certification helps.

**Questions:**

- From the paper, we can see that 7/9 space of this paper is used describe the introduction, related work, and method. Less than 2/9 of the space in the main paper are used to do experiment evaluation. Though there is some experimental analysis in the supplementary material, this is not adequate to well evaluate the proposed method and metric as posted in the weaknesses part.

I am not a researcher doing this research field but it really needs some efforts to further improve this paper.

---

> ### Author Response · Authors · 2023-11-21
>
> _Question: It seems that Theorem 1 is originally from (Fischer et al., 2021), not sure why put it in the main paper._
>
> Answer: Yes, we clearly state that Theorem 1 is from (Fischer et al., 2021). We added it for convenience and completeness, as it is central to our work, and we refer to it multiple times.
>
> _Question: A problem of introducing class hierarchy is that when applied to a new dataset, for example, the ADE20k dataset, a new class hierarchy graph should be prepared. Not sure how to make the proposed method universal to different segmentation datasets._
>
> Answer: For the purpose of our contribution, we assume a pre-defined class hierarchy. The class hierarchy is intrinsic to the class structure of each task/dataset. _AdaptiveCertify_ can be used with any hierarchy, and can therefore be applied to any segmentation task. Automatically creating hierarchies from a set of classes is a very different problem, which is beyond the scope of this paper. How the hierarchy is created has no effect on our certification algorithm.
>
> _Question: More datasets, with more than 19 classes, should be used to evaluate the performance of the proposed method._
>
> Reply: We agree that experiments on an additional dataset and investigating the effect of the number of classes will strengthen the submission. It is infeasible to run these experiments within the timeframe of the discussion, but we will add them for the final revision.
>
> _Question: I also think some segmentation results should be visualized to see in which cases the proposed adaptive hierarchical certification helps._
>
> Reply: Thank you for the suggestion, we agree and provide additional qualitative examples in the Appendix Section A.3 of our updated submission. We include examples where our method shows large improvements over the baseline in Figure 9, as well as randomly selected samples in Figures 10 and 11.
>
> _Question: From the paper, we can see that 7/9 space of this paper is used describe the introduction, related work, and method. Less than 2/9 of the space in the main paper are used to do experiment evaluation. Though there is some experimental analysis in the supplementary material, this is not adequate to well evaluate the proposed method and metric as posted in the weaknesses part._
>
> Reply: The core part of our method is that it guarantees the model’s robustness with respect to input perturbations. We therefore believe it is crucial to emphasize the method section and prove _SegCertify_’s soundness. As mentioned by the reviewer, there are additional experiments in the appendix, which explore different settings, parameters, and properties.
>
> _Question: Soundness: 2 fair_
>
> Reply: Why do you believe the method to be unsound? We propose a verification mechanism which proves robustness to adversarial perturbations, and prove this property in Section 4. Without detailed arguments, this statement is not helpful.

---

### Official Review · Reviewer_2yaR · 2023-10-31

**Soundness:** 2 fair
**Presentation:** 2 fair
**Contribution:** 1 poor
**Rating:** 3
**Confidence:** 3

**Summary:**

This paper introduces the concept of adaptive hierarchical certification for image semantic segmentation by mathematically formulating the problem and its adaptation to a pre-defined class hierarchy. This paper proposes ADAPTIVECERTIFY, the first adaptive hierarchical certification algorithm, which certifies the image pixels within different fine-to-coarse hierarchy levels.

**Strengths:**

1. The research problem is very important.
2. The paper is overall well-structured.

**Weaknesses:**

1. The technical contributions are not clear. There are some components combined together. What is the contribution of each component?
2. The experiments are insufficient. The authors should compare with more baselines on more datasets.

**Questions:**

1. Highlight the contribution of each component.
2. Provide more experimental results.

---

> ### Author Response · Authors · 2023-11-21
>
> Dear reviewer,
>
> Unfortunately, this review is very short and generic, which makes it difficult for us to provide a meaningful response. In our eyes, it is unsuitable to evaluate our paper in its current form. We kindly ask you to revise and improve your review.
>
> _Question: The technical contributions are not clear. There are some components combined together. What is the contribution of each component?_
>
> Answer: What do you mean by _“There are some components combined together?”_ It is unclear what components you are referring to. We discuss our exact contributions in Section 1 in the paragraph titled **Contributions**, and the overall method in Section 4.
>
> _Question: The experiments are insufficient. The authors should compare with more baselines on more datasets._
>
> Answer: Why do you think that the experiments are insufficient? There is only a single method for certification of semantic segmentation models, _SegCertify_, which we compare to. We are not aware of any additional baselines. Which methods are you asking to compare to?
>
> With regards to using more datasets, is there a specific quality you seek to explore using other datasets than Cityscapes and ACDC?
>
> _Question: Soundness: 2 fair_
>
> Answer: Why do you believe the method to be unsound? We propose a verification mechanism which proves robustness to adversarial perturbations, and prove this property in Section 4. Without detailed arguments, this statement is not helpful.

---

### Official Review · Reviewer_bW5R · 2023-10-31

**Soundness:** 2 fair
**Presentation:** 2 fair
**Contribution:** 2 fair
**Rating:** 6
**Confidence:** 1

**Summary:**

The work proposes an adaptive hierarchical certification method for segmentation. It adaptively relaxes the certification to a coarser level within the hierarchy, which helps lower the abstain rate and provides more semantic information. This problem is also mathematically formulated. Experiments also show that the proposed method beats the current state-of-the-art methods.

**Strengths:**

This paper is well-organized and presents a technically sound method to lower the abstain rate, which is also verified in the experiments.

**Weaknesses:**

I am not familiar with the area of certification for segmentation. When I read the preliminaries, the equations are not really that easy to follow. For example, in Eq.(1), what does IP means, and why do we use that. The authors should elaborate more clearly on that.

**Questions:**

Actually, I am sorry that I am not really an expert on this area, and at present I do not have enough time to learn the related knowledge from scratch. So I cannot give any professional comment. That would be great for us if the authors could give more detailed explanations on the theorems.

---

> ### Author Response · Authors · 2023-11-21
>
> $\mathbb{P}$ in Eq.1 means probability, as explained in the surrounding paragraph and all prior work on the topic. We give an overview of the most central concepts to understand our contribution and the related work in Section 2. However, it is beyond the scope of a background section in a 9-page conference paper to introduce the entire field of certification from the ground up.

---

### Official Review · Reviewer_dRWJ · 2023-11-01

**Soundness:** 2 fair
**Presentation:** 2 fair
**Contribution:** 2 fair
**Rating:** 3
**Confidence:** 3

**Summary:**

Tihis work presents an adaptive hierarchical certification method for semantic segmentation. To this end, the authors builds a hierarchical structure for the classes. Then, they reformulate the SegCertify with some changes for the semantic segmentation. In additon, the authors inroduce an evaluation metric. Experiments on Cityscapes and ACDC datasets partly verify the effectiveness of the proposed methods.

**Strengths:**

1. This work is easy to follow, since most of the techniques are based on previous works [Fischer et al.,2021][Lecuyer et al.,2019]
2. The proposed method shows better results than SegCertify [Fischer et al.,2021]

**Weaknesses:**

There are some key concerns:

1. Technical contributions

In fact, this work is largely based on SegCertify [Fischer et al.,2021]. The key differences are the hierarchical structure for semantic classes and the CIG metic for evaluation.  However, in my view, the proposed methods in section 4 are just a simple modification of SegCertify. There are no essential differences in theory. The new formulas are direct changes because of the different input shapes. As for the CIG, there are no insights why we should use that form. I agree that we need more robust certification methods for segmentation. However, the methods in this work is not a game-changer for this topic.

2.Insufficeint experiments

First, there are no comparisions with other segmentation methods. As a typical vision task, should the authors provide reuslts with the mIoU, mAP, etc? I suggest the authors add more comparisions with segmentation methods. Second, from the results in Tab.1, there are no significant differences in the performance (the gaps < 2%). They are too small. It is not convicning. Third, there are no enough examples to clarify the final conclusion. In fact, the authors only present the results with 100 images (If I misunderstand, corret me). Please list the results of the whole datasets. Finally, there are no visual examples for the failure cases. And the figure 1 is not enough.

3.Unclear details

There are many unclear details. For examples, what is the meaning of IP in Eq.1? What are the type I and type II errors? How do the SAMPLEPOSTERIORS and HSAMPLE operate?  What is the effect of using different hierarchical structure for semantic classes?

**Questions:**

Please see the weakness part.

---

> ### Author Response · Authors · 2023-11-21
>
> **(1) Technical Contribution**
>
> _This work is largely based on SegCertify [Fischer et al.,2021]_
>
> Our method is based on _SegCertify_. However. we don't have the same problem definition. We show that by introducing the concept of a semantic hierarchy and adapting a formal certification setup to it that we can abstain less while providing meaningful certified output. We introduce significant advances over the base method, which makes it applicable to the hierarchical certification setting and improves its CIG.. The major differences are: (i) We introduce and mathematically formulate the hierarchical certification for segmentation problem,  as discussed in Section 4.1, (ii) Due to the hierarchical nature, we have to sample from multiple hierarchy levels, as opposed to the flat hierarchy setup in _SegCertify_. We therefore introduce adaptive sampling in 4.3, particularly _HSample_ in Algorithm 2., (iii) Our implementation of _AdaptiveCertify_ in 4.4 is largely different from _SegCertify_ as we employ our novel functions that adapt to the hierarchical setup such as _GetComponentsLevels_ and _HSample_, (iv) All of these changes have to be done in a statistically sound manner, to ensure we compute sound guarantees. We show the soundness of _AdaptiveCertify_, which maintains the i.i.d assumptions in the sampling process, in Section 4.4.
>
> _As for the CIG, there are no insights why we should use that form_
>
> The insights behind using the CIG metric is that it accounts for the trades-off between class granularity and certified accuracy, as discussed in Section 4.6.
>
> **(2) Insufficient experiments**
>
> _Question: As a typical vision task, should the authors provide reuslts with the mIoU, mAP, etc?_
>
> Answer: This paper is not a typical computer vision paper. It discusses certification for a computer vision problem: segmentation. Therefore, only certified metrics (e.g., CIG) are relevant for comparison.
>
> _Second, from the results in Tab.1, there are no significant differences in the performance (the gaps < 2%)._
>
> The 0.02 improvement is the average over the test subset. For some, especially the challenging examples, it is higher reaching up to 0.04. Moreover, our improvement goes beyond CIG to also decreasing the abstain rate consistently (up to > 11%), and hence, providing more certified output (Table. 1).
> We have added visual examples highlighting the CIG performance increase and abstain rate decrease in the Appendix section A.3.
>
> _Third, there are no enough examples to clarify the final conclusion._
>
> We ran our experiments further to include 200 images from each dataset, and have updated Table. 1 accordingly. However, please not that certifying a subset of 100 images is common for certification work due to the high computational cost (Fischer et. al, 2021) and (Singh et. al, 2019).
>
> _Finally, there are no visual examples for the failure cases_
>
> Thank you for the suggestion! We provide additional examples for images where the samples are picked randomly in the Appendix Figures 10 and 11 under section A.3. On our test subset, we have not encountered failure cases where our certified information gain is lower than the baseline. Moreover, we discuss that our abstain rate will always be lower than the baseline in Section 4.5, as it also shows in the mentioned visual examples.
>
> **(3) Unclear details**
>
> _Question: For examples, what is the meaning of IP in Eq.1?_
>
> $\mathbb{P}$ in Eq.1 means probability, as explained in the surrounding paragraph and all prior work on the topic.
>
> _Question: What are the type I and type II errors?_
>
> Answer: Type I and type II errors correspond to the two standard error types of statistical hypothesis testing, which are basic concepts from the statistics literature. They are briefly mentioned in Sections 4.4 and 4.5. To further clarify, Type I and Type II errors occur when the null hypothesis is either wrongly rejected  or wrongly accepted, respectively. Our hypothesis test is conducted per component/pixel, and the null hypothesis is that the top vertex class probability is $< \tau$. There are two cases:
>
> *  (Type I error) Rejecting the null hypothesis while it is true: In our algorithm, this means that we certify a component, while its top class probability is $<\tau$. We bound the probability of this error by $\alpha$.
>
> *  (Type II error) Accepting the null hypothesis while it is false. In our algorithm, this means that we abstain from a component while in fact its top class probability is $\geq \tau$, which means it should have been certified.
>
> _Question: How do the SAMPLEPOSTERIORS and HSAMPLE operate?_
>
> SAMPLEPOSTERIORS computes $n$ samples of $f_\mathrm{seg}(x + \epsilon)$, as defined in Section 4.3.
> HSample is defined in Algorithm 2.
>
> _What is the effect of using different hierarchical structure for semantic classes?_
>
> Thank you for suggesting this! Given the limited time frame, we plan on including a different hierarchical structure in the final revision.

---

### Meta-Review · Area_Chair_PRsx · 2023-12-05

**Metareview:**

The paper presents an adaptive hierarchical certification method for semantic segmentation, with a focus on developing a hierarchical structure for classes and reformulating SegCertify for semantic segmentation. The reviews are consistent in their assessment, with all reviewers recommending rejection due to various concerns. Even though one reviewer gives the marginal acceptance but pointed out that he is not expert in this area. Thus, the paper is rated as not good enough by all reviewers, with ratings of 3 across the board.

**Justification For Why Not Higher Score:**

- Limited Technical Contributions: Reviewers point out that the work mainly builds upon existing methods, with only simple modifications proposed. This lack of substantial theoretical innovation limits the paper's contribution to the field.
- Insufficient Experimental Validation: The lack of comprehensive comparisons with other segmentation methods and the small performance improvements reported are major concerns. Additionally, the limited use of datasets and lack of extensive visual examples weaken the empirical validation.
- Lack of Clarity and Depth: Several reviewers note unclear details in the paper, including unclear explanations of key concepts and methodologies, which hampers the paper's overall comprehensibility and impact.
- Generalizability Concerns: The paper's approach appears to be dataset-specific and may not be easily adaptable to other datasets, which limits its applicability and relevance in the broader field.

**Justification For Why Not Lower Score:**

- Novelty in Approach: The introduction of a hierarchical certification method for semantic segmentation is recognized as a novel approach. The paper takes initial steps in exploring this area, which is a positive aspect.
- Good Structural Presentation: Despite concerns about depth, the paper is acknowledged for its well-structured presentation, which aids in understanding the proposed methodology and its background.
- Some Positive Experimental Results: The paper does show that the proposed approach achieves better results than SegCertify under the CIG metric, providing some evidence of its potential effectiveness.

---

### Decision · Program_Chairs · 2024-01-16

Reject